# Artificial Grooming during Early Life could Boost the Activity and Human Affinity of Holstein Female Calves

**DOI:** 10.3390/ani10020302

**Published:** 2020-02-13

**Authors:** Congcong Li, Jian Wang, Shuang Jin, Xianhong Gu

**Affiliations:** State Key Laboratory of Animal Nutrition, Institute of Animal Sciences, Chinese Academy of Agricultural Sciences, Beijing 100193, China; congcongli1988@sina.com (C.L.); wangjian_1884@163.com (J.W.); shuangjinjs@163.com (S.J.)

**Keywords:** activity, artificial grooming, cow–calf separation, Holstein female calf, human affinity, starter diet

## Abstract

**Simple Summary:**

Due to early cow–calf separation in modern farms, maternal care, such as licking and grooming, is greatly limited for newborn calves. Physically imitating the maternal licking with manual brushing, termed artificial grooming, during an early age might substitute the role of cows. The behavioral response of dairy calves with artificial grooming treatment was investigated. We found beneficial effects of artificial grooming on the activity and human–animal bonding of dairy calves. The starter ingestion might be advanced by the artificial grooming and might contribute to a less stressful weaning process. Mechanization of this process is promising in the future calf management and thus improving the welfare status of dairy calves.

**Abstract:**

Early cow–calf separation management induced various welfare problems for dairy calves. We mimicked the maternal licking by manually brushing right after the Holstein female calves were born and during their first week of life, termed artificial grooming (AG). The behavior of these treated calves (AG, *n* = 17) was compared with the calves without artificial grooming (Con, *n* = 16) during daily behavioral observation around evening milk feeding and in the open field test (OFT) and novel human test (NHT). The number of calves ingesting starter on day six was recorded. The AG calves were observed to be more active and perform more oral behavior compared with the Con calves around evening milk feeding. In the OFT and NHT, the AG calves were again more active than the Con calves. Moreover, the AG calves tended to be less cautious and had more human interactions than the Con calves in the NHT. There tended to be a higher percentage of AG calves ingesting starter on day 6. In conclusion, artificial grooming during early life could boost the activity and the human affinity of female calves and it might advance their starter diet ingestion.

## 1. Introduction

For most mammalian species, the licking of the neonate is a common behavior pattern to remove fetal membranes, dry coat, keep environment hygienic, assure infant survival, and form mother–infant bond [1]. In terms of dairy cows, right after calving, the predominant behavior is the newborn calf licking, occupying 30% to 50% of the first hour postpartum [1]. The early presence of the dams has been found to increase motor activity [2], social behavior [3], and later maternal behavior [4] of calves. Under natural conditions, cow–calf pairs remain together for around 6–8 months, during which calves could naturally get through the time of weaning [3]. However, in modern industrial dairy farms, early separation of cow and calf is widely implemented for the sake of economic, health, and compassionate reasons [3]. Artificial rearing is considered by most producers to provide fine supervision of calves and reduce the distress of cow–calf separation at a later age [3]. On the contrary, some others believe that early separation is emotionally stressful for the calves and cows, negatively affecting their health [5]. The management of immediate cow–calf separation postpartum deprives calves of being licked, which prolongs the period of calves being wet and unaccompanied under a very stressful situation. This management is greatly criticized from an animal welfare point of view [6]. The effects of early cow–calf separation on the cow and calf health were systematically reviewed by Beaver et al. [7]. The same group reported that longer cow–calf contact typically had positive long-term effects on calves in the aspect of behavior and growth performance [8].

To establish a sustainable and environmentally friendly production system while at the same time focusing on animal health and welfare, organic farming has been steadily increasing in Norway [9]. In their organic dairy farms, it is strictly ruled that calves must be allowed to suckle from dams for at least three days postpartum [9]. This practice of keeping calves with their mothers is applicable as the number of cows per farm is relatively small, from 15 to 18 [10,11]. However, with the rapid growth of the Chinese dairy industry since 2008, dairy farms with herd size under 50 no longer exist, and 61.4% of dairy farms had over 100 cows in 2018 (White Paper of China Dairy, 2019). Keeping calves together with dams is a challenging management practice for large dairy farms, however, the welfare of the significant number of newborn calves should not be neglected. Numerous studies have proven that massage is beneficial for preterm infants across the US, Europe, and Asia [12,13]. Moderate pressure massage has pacifying or stress-reducing effects [12]. It could increase vagal activity and thus lead to increased gastric motility and weight gain of preterm infants [14]. Chen et al. [15] reported that baby massage at an early stage after birth could ameliorate neonatal jaundice. Therefore, we hypothesize that the mother’s licking of the calves is similar to the baby massage process and can be mimicked artificially. Haines and Godden conducted 15 min of vigorous physical and verbal stimulation, termed artificial mothering, within 1 to 2 h after birth to examine its effect on IgG passive transfer in dairy calves, and no significant effect was observed [16]. Yet, the behavioral response of calves with physical and verbal stimulation was not investigated. In the current study, we only conducted the physical stimulation as the verbal stimulation of cows was difficult to mimic and this procedure was termed artificial grooming. The artificial grooming was conducted by humans, and this could be taken as a kind of positive human handling as well. Early positive handling of calves by human was reported to have beneficial effect on their socializing behavior and response to humans [17,18]. It is hypothesized that the activity level might be elevated by the artificial grooming treatment and their response to human presence might also be affected. It is also interesting to picture their general behavioral pattern with the treatment. Therefore, we examined the effect of artificial grooming on the activity, human affinity, and starter diet ingestion status of calves by focally observing their behavior in daily rearing and arena tests. We expect that artificial grooming during early life could positively influence the behavioral pattern of calves and thereby improve their welfare status.

## 2. Materials and Methods

The experimental Animal Care Committee of the Institute of Animal Sciences, Chinese Academy of Agricultural Sciences has approved the conduction of the current experiment under the accession number of IAS2018-6.

### 2.1. Animals and Husbandry Management

This study was carried out at a dairy farm named Hebei Shounong Modern Agricultural Technology Co. Ltd., in the Hebei province of China, in October 2018. Thirty-three Holstein calves born from primiparous cows were selected and sequentially allocated to control (Con; *n* = 16) or artificial grooming (AG; *n* = 17) groups and balanced by body weight (mean ± SD; Con 37.8 ± 2.7 kg, AG 36.8 ± 2.2 kg). As primiparous cows have a higher chance of difficulty in calving, manual assistance was conducted when there was no noticeable progress 60 min after the appearance of the allantois and feet. Unusually, most of the calves were delivered with manual assistance, which might be due to the general higher body condition score (mean ± SD; 4.0 ± 0.3) of the cows. Therefore, the calves with calving-ease score between 2 to 4 (calving difficulty was evaluated based on a 5-point calving-ease score, with 1 = unassisted, 2 = easy pull, 3 = moderate pull, 4 = very hard pull, and 5 = Caesarian section) were used in the present study [16], and the calving difficulty level was balanced between groups (mean ± SD; Con 2.6 ± 0.7, AG 2.7 ± 0.7). Immediately after birth, calves were removed from their dams and ear-tagged, umbilical cord disinfected, and colostrum fed in a transit house. After being kept in the transit house for 6 to 12 h, they were placed individually in 1.5 m × 2 m plastic hutches with a 1.5 m × 1.25 m exercise pen in front of each hutch. Hutches were aligned in two rows with 16 hutches in each row, and the location of hutches used for each treatment was balanced. Shade structures were provided for all the hutches and exercise areas. Bedding for calves was fine sand covered on top with straw. Colostrum (≥50 mg of IgG/mL) of 10% body weight was given to calves through an esophageal tube within 1 h after birth (day 0). Blood samples were collected from the jugular vein 24 h after the colostrum feeding. The protein level of serum was analyzed using a portable hand-held refractometer (THE01516, Yieryi). The serum protein level of calves included in the study was all above 6.0 g/dL. From day 1 onwards, calves were open-bucket fed three times with 2 L whole cow milk per day at 08:30, 14:30, and 20:30. Warm water was provided around 10 min after finishing milk feeding for all calves each day. Starter diet pellets (Beijing Sanyuan Breeding Technology Co., Ltd., meeting the nutrient requirements of the national research council (NRC); crude protein, fiber, fat, and ash contents were around 20%, 8%, 2%, and 15%, respectively) were provided for all calves starting from day 3 [19]. An experienced veterinarian supervised all calves daily. One dehydrated calf from the Con group and one front-leg disabled calf from the AG group were excluded from the experiment, resulting in the aforementioned numbers of experimental calves. Eventually, only clinically healthy calves were included in the present study.

### 2.2. Experimental Treatments

The calves were included in the experiment from immediately after birth until seven days of age. Immediately after birth, the Con calves were separated from their dams and vigorously dried within 1–2 min by a towel and thereafter left alone. Contact with the Con calves, including physical, tactile, and verbal stimulation were minimized during the whole experimental period. Besides vigorously drying, licking of the AG calves by their dams was mimicked by manually brushing for 30 min after birth (day 0) and before colostrum feeding. The body parts of the neck, brisket, shoulder, back, and thigh were groomed, and each body part took up 6 min. The area of rump and pin bones were avoided because we found that these are sensitive areas when being touched. It is difficult to quantify the intensity of the artificial grooming, but we would like to describe this practice with “moderate pressure grooming”. The 30 min artificial grooming was conducted in the first place in a transit house where the calves were ear-tagged, umbilical cord disinfected, and colostrum fed sequentially after the treatment. The time of ear-tagging, umbilical cord disinfection, and colostrum feeding of all calves was around 0.5 to 1 h after birth. During the first half-hour after the birth of the Con calves, there was human (the same person who conducted the AG treatment) existence within a 1-m radius. Afterward, the calves were transferred to the individual hutches. Artificial grooming was continued for the AG calves from day 1 to day 6 by conducting 5 min of brush grooming 1 to 2 h after morning (08:30) and noon (14:30) milk feeding. The 5 min of brush grooming was mainly focused on the neck and back of the calves. As the location of hutches used by calves was balanced between treatment, calves with or without artificial grooming were always neighbors. Therefore, when the treatment calves were being brush groomed, the neighbor control calves could always see the presence of the human but were not groomed. Due to the long distance between the calving barn and the hutch area, the 30 min of artificial grooming right after the birth of all calves was conducted by one person, and another person performed the two daily times of 5 min grooming at the hutch area.

### 2.3. Daily Behavioral Observation

Eight supervision cameras (Hikvision Digital Technology Co., Ltd., Hangzhou, China) were placed 3 m away in front of the hutches, and each camera was responsible for four hutches. The behavior of all calves was continuously filmed during the periods of 08:00 to 12:00, 13:00 to 17:00, and 19:00 to 22:00 on all experimental days. As there was the least external interference during the evening, the behavior of calves during the evening milk feeding (20:30) as well as the 10 min before and after that on days 4 and 6 was focally observed. The duration and bouts of behaviors such as lying, standing, walking, jump-running, object sniffing, object sucking and licking (object S/L), object rubbing, butting, self-grooming, and eliminating were recorded. The ethogram of all behaviors observed can be found in Table 1. In addition, one-zero sampling of starter eating behavior was used to check the percentage of calves ingesting starter diet during all the periods recorded on day 6. If the calves were observed to ingesting the starter diet and chewing no less than 10 s, the calves were considered starter diet consumers. The behavioral observer was blinded to the treatments during focal and one-zero samplings.

### 2.4. Behavioral Tests

On day 7, one hour after morning milk feeding, an open field test (OFT) of 5 min session time was conducted for each calf in an empty and unfamiliar testing arena measuring 6 m × 7 m. The walls of the testing arena were fixed with dark brown waterproof cloth (1.2 m height) and the ground was covered with fine sand. The testing arena was 16 m away from the center of the experimental hutch area. To start the OFT, a person, who had no contact with the calves, gently led one testing calf walking to the testing arena and placed the calf at the arena center. After positioning the calf, the person quickly left the arena, and the calf was alone without any visual and auditory interference. Followed with the OFT, another 5 min session of the novel human test (NHT) was carried out. To minimize the external stress, a novel person (who was different from the leading person and never had any contact with the calves) entered the arena and stood at the center right after the OFT with the calf where it was in the arena (none of the calves was at the arena center). During the 5 min session of NHT, the person stood still and had no eye contact with the testing calf. It was always the same novel person involved in the NHT. A supervision camera (Hangzhou Hikvision Digital Technology Co., Ltd.) was mounted 4 m above the testing arena to have full supervision of the area. Videos were recorded for further behavioral observation. To obtain the most accurate and detailed results, we used focal sampling for the behavioral observation of calves in the OFT and NHT. The duration and bouts of behaviors, including standing, walking, jump-running, object sniffing, self-grooming, and eliminating, were recorded in the OFT. Other behaviors such as staring at the person while standing (Stand-S), human sniffing, human rubbing or licking (human R/L), and the latency the calves took in walking towards human (L-WtoH) were also recorded in the NHT. Table 1 lists the ethogram of behaviors observed in the arena tests. During the focal sampling, the behavioral observer had no information about the treatment of the calves.

### 2.5. Statistical Analysis

All data were analyzed with the JMP Pro 14 software (SAS Institute Inc., Cary, NC, USA). The duration of all behaviors observed around the daily evening milk feeding and arena tests was expressed as seconds. To obtain normal distributions, the behavioral duration data were log (x + 1) transformed, and the behavioral bouts were square-root transformed. Repeated measure ANOVAs were used to examine the effects of treatment, day, period, and their interactions on each behavior duration and bouts of calves for the 10 min before and after evening milk feeding (20:30). The same analytic method was used for the period of during evening milk feeding. The two observational days (days 4 and 6) were considered repeated measures. One-way ANOVA was performed to test the effect of treatment on each behavior duration of calves in the OFT and NHT. The nonparametric Kruskal–Wallis test was used to assess the treatment effect on each behavior bout of calves in the OFT and NHT. Chi-squared tests were conducted to evaluate the treatment effect regarding the percentage of calves ingesting starter diet on day 6. *p* ≤ 0.05 is taken as significantly different. A tendency of difference was defined when 0.05 < *p* ≤ 0.10 was observed.

## 3. Results

### 3.1. Behavior around Evening Milk Feeding

The AG calves spent more time and performed more bouts of walking (Duration *p* = 0.005, Bouts *p* = 0.023) compared with the Con calves before evening milk feeding (Table 2 and Table 3) whereas no difference was observed after evening milk feeding (Period*Treatment: Duration *p* = 0.023, Bouts *p* = 0.029). Before and after evening milk feeding, the AG calves spent more time and performed more bouts of jump-running (Duration *p* = 0.014, Bouts *p* = 0.010) and object S/L (Duration *p* = 0.023, Bouts *p* = 0.017) behavior compared with the Con calves (Table 2 and Table 3). After evening milk feeding, AG calves spent more time butting compared with the Con calves (*p* = 0.012) (Table 2). No difference in the duration and bouts of butting behavior was observed before evening milk feeding (*p* > 0.05) (Table 2 and Table 3).

There are period effects for the duration and bouts of all behaviors observed except for the eliminating behavior (Table 2 and Table 3). In detail, the lying behavior was more performed in terms of duration (*p* < 0.0001) and bouts (*p* < 0.0001) before evening milk feeding than after evening milk feeding (Table 2 and Table 3). Compared with the period of before evening milk feeding, the calves had longer duration and more bouts in performing standing (Duration *p* < 0.0001, Bouts *p* < 0.0001), walking (Duration *p* = 0.001, Bouts *p* = 0.002), jump-running (Duration *p* < 0.0001, Bouts *p* < 0.0001), butting (Duration *p* < 0.0001, Bouts *p* < 0.0001), self-grooming (Duration *p* < 0.0001, Bouts *p* < 0.0001), object sniffing (Duration *p* < 0.0001, Bouts *p* < 0.0001), object rubbing (Duration *p* = 0.001, Bouts *p* = 0.001), and object S/L (Duration *p* < 0.001, Bouts *p* < 0.0001) after evening milk feeding (Table 2 and Table 3).

On day 4, the calves had longer duration (*p* = 0.024) and more bouts (*p* = 0.049) in performing walking behavior than they did on day 6 (Table 2 and Table 3). The calves tended to have longer standing (*p* = 0.064) and butting (*p* = 0.098) durations as well as have significantly more standing (*p* = 0.042) and butting (*p* = 0.018) bouts on day 4 compared with those on day 6 (Table 2 and Table 3). Compared with day 4, there was a tendency of more object sniffing (Duration *p* = 0.095) and object S/L (Duration *p* = 0.054, Bouts *p* = 0.057) on day 6 (Table 2 and Table 3). On day 4, the calves conducted significantly more self-grooming bouts after evening milk feeding compared with before evening milk feeding; the same, but less significant, difference was observed on day 6 (Day *p* = 0.026, Day*Period *p* = 0.003) (Table 3). In terms of the behaviors during evening milk feeding, we found no difference in the duration and bouts of all behaviors in the calves (*p* > 0.05, data not shown). The measure of seconds taken to finish the milk provided in the evening had no difference between the treatments. (mean ± SEM; Day 4: Con 189.1 ± 28.5, AG 183.4 ± 23.5; Day 6: Con 188.9 ± 30.7, AG 199.1 ± 29.0; Treatment *p* = 0.903, Day *p* = 0.526, Treatment*Day *p* = 0.574)

### 3.2. Behavior in the Open Field Test

Figure 1 exhibits the behavioral results of the calves in the OFT. The artificial grooming treatment again tended to increase the duration (*p* = 0.079) and number of bouts (*p* = 0.090) of the jump-running behavior of the calves. No other differences in the behavior conducted in the OFT between the Con and AG calves were observed (*p* > 0.10).

### 3.3. Behavior in the Novel Human Test

The results of the NHT of the calves are shown in Figure 2. The staring at the person while standing behavior of calves tended to decrease (Duration *p* = 0.100, Bouts *p* = 0.059), whereas the rubbing and licking human behavior tended to increase (Duration *p* = 0.076, Bouts *p* = 0.080) with the treatment of artificial grooming. The duration and number of bouts of jump-running behavior were significantly greater in the AG calves than in the Con calves (Duration *p* = 0.012, Bouts *p* = 0.027). In the NHT, the latency the calves took to walk to the human was (mean ± SEM) 178.6 ± 35.8 s and 92.2 ± 29.9 s for the Con and AG calves, respectively (*p* = 0.055) (due to the figure design consideration, this information is not included in Figure 2 or in other tables or figures). The duration and bouts of other behaviors in the NHT were similar between the two treatments (*p* > 0.10).

### 3.4. Percentage of Calves Ingesting Starter Diet

On day 6, there was 64.7% of AG and 37.5% of Con calves ingesting the starter diet (*p* = 0.054, Figure 3).

## 4. Discussion

As far as we know, this is the first study to investigate how artificial grooming right after birth and during the first week affects dairy calves in the aspect of behavior. In the current study, artificial grooming refers to mimicking maternal licking by manually brushing the body of calves. By doing the daily behavioral observation and arena tests, we found that the artificial grooming treatment revealed its effect on calves mainly in the aspect of physical and oral activity, as well as human affinity.

### 4.1. General Activity and Playing

Artificial grooming during early life increased the general activity of Holstein female calves, which is well-proven by the increased jump-running behavior in various behavioral observation sessions. The artificial grooming functioned similarly to the maternal licking of newborn calves. For the majority of mammalian species, the predominant maternal behavior following parturition is licking [1]. Such licking could dry the coat and reduce heat loss as well as stimulate breathing and blood circulation, which are contributing factors for a higher level of activity of infants [21,22]. Townsend and Bailey [23] observed a higher activity level of the first fawn compared to its siblings, and this higher motor development might derive from the stronger maternal licking performance after the first birth. The maternal licking might have declined in vigor following the siblings’ birth. The experimental calves that had the artificial grooming treatment exhibited a higher activity level, mainly reflected in more jump-running behavior. The jump-running behavior is widely considered as a playful behavior of calves and is a good indicator of welfare in young mammals [24,25]. Valníčková et al. [25] pointed out that individually-housed calves are deprived of natural levels of play as they have lower spontaneous play behavior. Therefore, it is quite encouraging that calves with artificial grooming treatments are more playfully active compared with the control calves, indicating a general better welfare.

### 4.2. Oral Activity

The calves in our experiment received limited amount of milk daily, and this restricted feeding has been reported to induce hunger in the study of Vieira et al. [26]. They indicated that milk-restricted calves were more active and performed more unrewarded visits to feeders than calves fed ad libitum. Under this unsatisfied milk intake, the calves that had artificial grooming treatment exhibited more object sucking and licking behavior compared with the control calves before and after evening milk feeding. Longer duration of butting behavior was also observed in the artificially groomed calves after evening milk feeding. This highly motivated behavior in craving for milk of the artificially groomed calves might be due to their reduced fear with twice daily human interactions. It has been evidenced that individually-housed calves experience higher levels of fear than group-housed calves [27]. The calves might gain more confidence with the daily human interaction. In addition, all the calves in our study were open-bucket fed and never had a chance to perform natural milk-sucking behavior. Object sucking is believed to develop when there is a lack of opportunity to suckle or unsatisfied suckling motivation [28]. There were increasing day effects on oral-activity-related behaviors such as object sniffing and object S/L. Calves at an older age (day 6) have a higher energy requirement than at an early age (day 4), but the milk amount we provided was the same from day 1 to day 7. Therefore, more milk should be provided in releasing their unrewarded sucking or licking motivation. In terms of our experimental treatment, we hypothesize that the artificial grooming treatment probably enhanced the motivation of calves to perform natural sucking behavior by potentially reducing their fear level.

### 4.3. Starter Diet Ingestion

In the current study, the percentage of artificially groomed calves ingesting starter diet on day 6 tended to be higher compared with that of the control calves. Early consumption of the starter diet is critical for the development of an active and functioning rumen, which will allow early weaning of calves and thus have positive effects on the following growth and production performance [19,29]. Usually, calves start consuming small amounts of solid feed around 14 days of age [19]. Neave et al. [30,31] observed a large variation in the age (range from 4 to 41 days of age in the study in 2018; range from 18 to 75 days of age in the study in 2019) when calves first found and began to consume starter. Some of our experimental calves started to consume the starter diet as early as 6 days of age, and the artificial grooming group tended to have a higher proportion of starter-diet-consuming calves. There is evidence that calves that were more exploratorily active began to consume starter at an earlier age, showed greater starter dry matter intake, and had higher overall average daily weight gain [30]. There is also a positive association between the level of exploratory activity and preweaning of calves [31]. Therefore, it seems likely that artificial grooming treatment might accelerate the solid food intake of calves by increasing their activity level.

### 4.4. Human Affinity

As expected, Holstein calves that had artificial grooming treatment showed to be less vigilant and more playful in the presence of a novel person and ultimately had more interactions with the person in the NHT. A good human–animal relationship in farms is important as it can reduce the stress responses of animals to routine management practices and thus improve animal welfare status. Positive handling of dairy calves early in the rearing period was reported to have beneficial effects on the response of calves to human interactions [32]. Lürzel et al. [33] performed 42 min of gentle interactions during the first 14 days of life on female Holstein–Friesian calves, and the gentle interactions were effective in fear reduction towards humans in both the short term and long term after weaning. Similar studies were also conducted with piglets and positive effects have been observed such as increased handler interactions [34] and reduced fearfulness in a novel environment [35]. Stroking different parts of the body had various effects on the human response of cows, and the ventral part of the neck was much accepted [36]. In the current study, the artificial grooming group had in total 90 min of manual brushing with the presence of a person during the first week of their life. The response of Holstein female calves to the novel person in the NHT agreed with the studies referred to.

## 5. Conclusions

In the present study, the procedure of physical stimulation by manual brushing was termed as artificial grooming. Carrying out artificial grooming during early life might reduce the fear level of calves and thus result in increased playful activity, more positive human response, more oral behavior, and the potential of advancing starter ingestion of Holstein female calves. It is worth pointing out that the calves in the present study received a limited amount of milk, which could have influenced their behavior to a certain extent. The obvious benefits of artificial grooming suggest that this procedure should be considered in calf management in practice. For the sake of labor, this procedure can be mechanized, and further studies should be done, such as the investigation of grooming intensity, frequency, and detailed calf responses.

## Figures and Tables

**Figure 1 animals-10-00302-f001:**
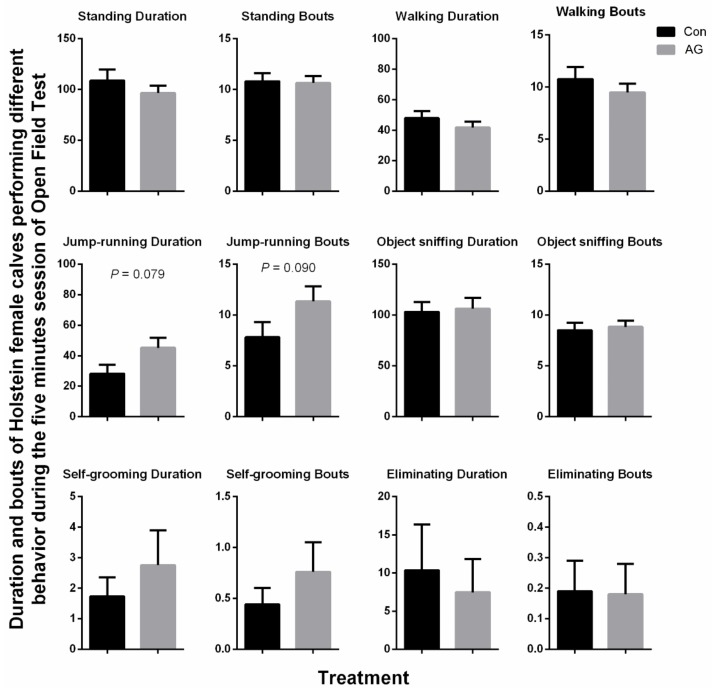
The effect of artificial grooming on the behavior of Holstein female calves in the open field test on day 7 (s). Effect direction is provided when the mean effect is *p* ≤ 0.10. Data are presented as means with standard error bars of SEM. Control group (Con), *n* = 16; Artificial grooming group (AG), *n* = 17.

**Figure 2 animals-10-00302-f002:**
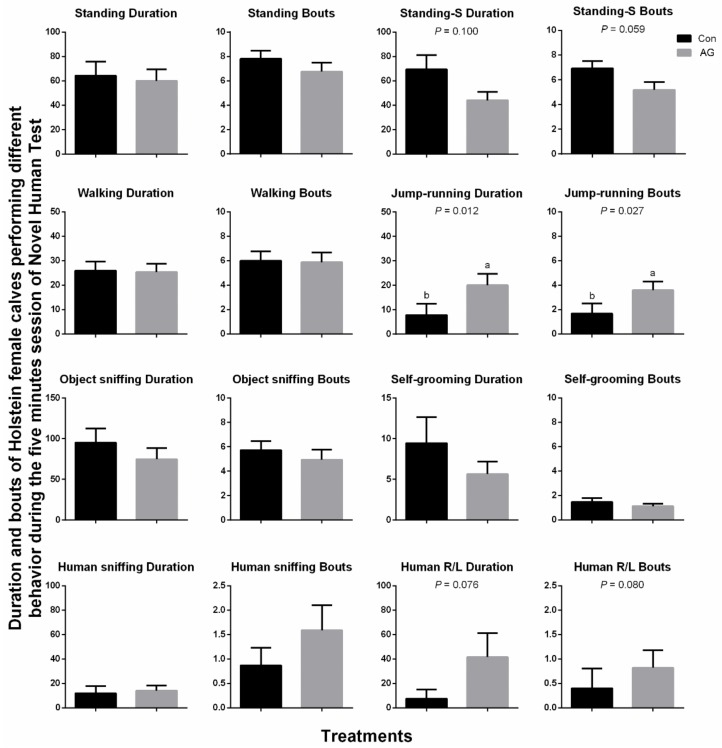
The effect of artificial grooming on the behavior of Holstein female calves in the novel human test on day 7 (s). Standing-S = staring at the human while standing; Human R/L = human rubbing or licking. Effect direction is provided when the mean effect is *p* ≤ 0.10. ^a,b^ Different superscript letters indicate a significant difference between groups. Data are presented as means with standard error bars of SEM. Control group (Con), *n* = 16; Artificial grooming group (AG), *n* = 17.

**Figure 3 animals-10-00302-f003:**
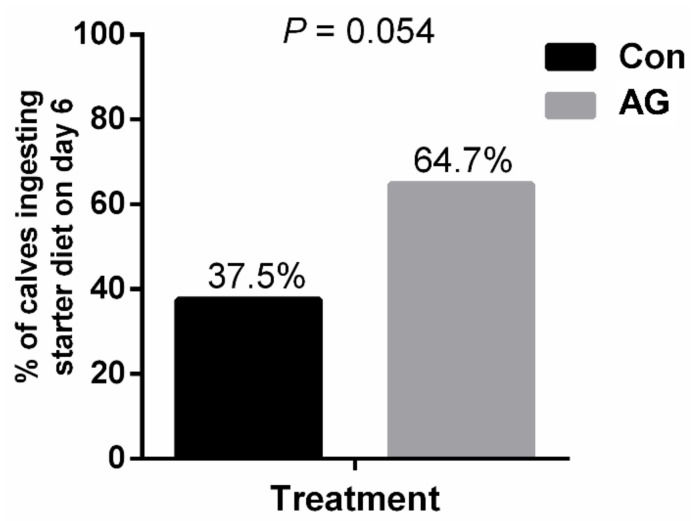
The effect of artificial grooming on the percentage of Holstein female calves ingesting starter diet on day 6. Effect direction is provided when the mean effect is *p* ≤ 0.10. Data are presented as percentages.

**Table 1 animals-10-00302-t001:** The ethogram of behavior observed around evening milk feeding and during open field test and novel human test of Holstein female calves (partially based on Duve and Jensen, [20]).

Session	Behavior	Definition
Around evening milk feeding time	Lying	Body in contact with bedding, or in the process of lying down or standing up
	Standing	Upright with 3 to 4 hooves in contact with bedding, supporting the weight of the body without quick head movement and any oral activities
	Object sniffing	Muzzle in physical contact or close with any object in the hutch and exercise area
	Walking	Slowly moving forward or backward without any head activities
	Self-grooming	Licking (tongue in contact with fur) or scratching (hind leg lifted from the ground and hoof in contact with fur) any part of own body or head
	Butting	Quickly force pushing the head against any object or in the air
	Jump-running	Both forelegs (hind legs) are lifted from the ground and stretched forward (backward) followed with continuous running in circles, back and forth or with constant changes in direction; both gallop and trot are included
	Object S/L	Sucking or licking any object in the hutch and exercise area
	Eliminating	Excretion and/or urination
	Object rubbing	Forehead moving up and down rhythmically against any object in the hutch and exercise area
Open field test	Standing	See above
	Object sniffing	Muzzle in physical contact or close with any object in the arena
	Walking	See above
	Self-grooming	See above
	Eliminating	See above
Novel human test	Standing	See above
	Standing-S	Staring at the human while standing
	Object sniffing	Muzzle in physical contact or close with any object in the arena
	Human sniffing	Muzzle in physical contact or close with the human
	Walking	See above
	Self-grooming	See above
	Jump-running	See above
	Human R/L	Forehead moving up and down rhythmically against the human or licking the human

Object S/L = object sucking and/or licking; Human R/L = human rubbing and/or licking.

**Table 2 animals-10-00302-t002:** The effect of artificial grooming on the behavior duration (s) of Holstein female calves before and after evening milking feeding on days 4 and 6.

Behavior	Per	Day 4		Day 6		SEM	*p*-Value					
		Con	AG	Con	AG		Per	D	Tr	Per*D	Per*Tr	D*Tr
Lying	Be-EMF	303.3	288.5	296.3	238.0	28.9	***	ns	ns	ns	ns	ns
	Af-EMF	23.1	26.5	52.0	41.3	12.6						
Standing	Be-EMF	214.1	207.1	170.3	203.9	19.6	***	†	ns	ns	ns	ns
	Af-EMF	320.7	297.7	274.4	247.8	13.4						
Walking	Be-EMF	16.9 ^b^	27.1 ^a^	11.4 ^b^	26.6 ^a^	2.1	**	*	*	ns	*	ns
	Af-EMF	31.6	34.3	21.8	18.8	2.0						
Jump-running	Be-EMF	1.8 ^b^	12.2 ^a^	3.3 ^b^	10.4 ^a^	1.8	***	ns	*	**	ns	ns
	Af-EMF	24.6 ^b^	59.9 ^a^	18.2^b^	27.0 ^a^	4.4						
Object sniffing	Be-EMF	37.7	38.5	76.4	86.7	9.5	***	†	ns	ns	ns	ns
	Af-EMF	114.4	103.4	129.4	109.8	7.4						
Object rubbing	Be-EMF	0.0	0.3	4.3	0.0	1.0	**	ns	ns	ns	ns	ns
	Af-EMF	6.4	2.8	5.8	5.8	1.5						
Object S/L	Be-EMF	0.0 ^b^	3.6 ^a^	0.1 ^b^	3.4 ^a^	0.8	***	†	*	ns	ns	ns
	Af-EMF	2.1 ^b^	6.1 ^a^	26.8 ^b^	30.8 ^a^	6.8						
Butting	Be-EMF	0.4	0.0	0.0	0.0	0.1	***	†	ns	ns	*	ns
	Af-EMF	1.2 ^b^	4.9 ^a^	0.5 ^b^	2.2 ^a^	1.4						
Self-grooming	Be-EMF	11.1	19.5	27.9	13.1	3.4	***	ns	ns	**	ns	ns
	Af-EMF	65.9	66.6	37.9	53.8	5.4						
Eliminating	Be-EMF	14.7	3.2	9.9	14.4	3.1	ns	ns	ns	ns	ns	ns
	Af-EMF	10.0	5.3	16.1	9.6	3.1						

Per = period; SEM = standard error of the means; D = day; Tr = treatment; Be-EMF = before evening milk feeding; Af-EMF = after evening milk feeding; Object S/L = object sucking and/or licking. ns = no significant difference; † 0.05 < *p* ≤ 0.10; * *p* ≤ 0.05; ** *p* ≤ 0.01; *** *p* ≤ 0.001. ^a,b^ Different superscript letters in the same row indicate a significant difference between treatments. Data are presented as means. Control group (Con), *n* = 16; Artificial grooming group (AG), *n* = 17.

**Table 3 animals-10-00302-t003:** The effect of artificial grooming on the behavior bouts of Holstein female calves before and after evening milking feeding on days 4 and 6.

Behavior	Per	Day 4			Day 6		SEM	*p*-Value					
		Con	AG		Con	AG		Per	D	Tr	Per*D	Per*Tr	D*Tr
Lying	Be-EMF	0.8	0.9		1.2	0.7	0.1	***	ns	ns	ns	ns	ns
	Af-EMF	0.6	0.1		0.2	0.1	0.1						
Standing	Be-EMF	8.4	10.5		7.2	11.5	1.0	***	*	ns	ns	ns	ns
	Af-EMF	22.0	23.6		17.8	17.5	1.0						
Walking	Be-EMF	5.1 ^b^	6.5 ^a^		3.3 ^b^	6.7 ^a^	0.6	**	*	ns	ns	*	ns
	Af-EMF	8.1	8.5		6.9	5.3	0.5						
Jump-running	Be-EMF	0.6 ^b^	2.9 ^a^		1.2 ^b^	3.3 ^a^	0.5	***	ns	**	*	ns	ns
	Af-EMF	7.4 ^b^	13.7 ^a^		5.5 ^b^	8.3 ^a^	1.2						
Object sniffing	Be-EMF	3.8	3.9		3.8	6.0	0.6	***	ns	ns	ns	ns	ns
	Af-EMF	10.6	10.8		9.8	9.7	0.6						
Object rubbing	Be-EMF	0.0	0.1		0.1	0.0	0.0	**	ns	ns	ns	ns	ns
	Af-EMF	0.5	0.2		0.5	0.3	0.1						
Object S/L	Be-EMF	0.0 ^b^	0.1 ^a^		0.1 ^b^	0.3 ^a^	0.1	***	†	*	ns	ns	ns
	Af-EMF	0.3 ^b^	0.7 ^a^		0.6 ^b^	1.7 ^a^	0.2						
Butting	Be-EMF	0.3	0.0		0.0	0.0	0.0	***	*	ns	ns	†	ns
	Af-EMF	1.0	3.6		0.4	1.6	0.2						
Self-grooming	Be-EMF	1.6	1.5		2.5	1.7	0.2	***	*	ns	**	ns	ns
	Af-EMF	7.5	7.5		4.1	5.2	0.5						
Eliminating	Be-EMF	0.3	0.1		0.1	0.2	0.0	ns	ns	ns	ns	ns	ns
	Af-EMF	0.4	0.1		0.3	0.3	0.1						

Per = period; SEM = standard error of the mean; D = day; Tr = treatment; Be-EMF = before evening milk feeding; Af-EMF = after evening milk feeding; Object S/L = object sucking and/or licking. ns = no significant difference; † 0.05 < *p* ≤ 0.10; * *p* ≤ 0.05; ** *p* ≤ 0.01; *** *p* ≤ 0.001. ^a,b^ Different superscript letters in the same row indicate a significant difference between treatments. Data are presented as means. Control group (Con), *n* = 16; Artificial grooming group (AG), *n* = 17.

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
