# Peer review of "Artificial Grooming during Early Life could Boost the Activity and Human Affinity of Holstein Female Calves"

_animals, 2020, doi:10.3390/ani10020302_

Round 1
Reviewer 1 Report
This study provided new insights on effects of artificial brushing to calves on their activity and ingestion of starter diet, which have been investigated from the only view of handling easiness.
Minor revisions are needed as follows;
Line 138 and 164: Focal sampling and continuous sampling are not identical. Then, you must write more precisely sampling method. Line 137 and 178: The behavior of calves was observed before (10 min), during, and after (10 min) the evening milk feeding. How many minutes is the evening milk feeding? The time budget of the evening milk feeding must be written. And there was no results on the behavior of calves during the evening milk feeding. Table 1: What does the word “tough” using in Definition of Object S/L and Human R/L mean? The word “licking” is used in descriptions of Object S/L and Human R/L on the margin. What is relationship between “tough” and “licking”? Line 152: Walls and floor structure of the testing arena should be written, because the structure must influence the behavior of calves. Line 231-232: Table 3 shows that bouts of self-grooming of Day 6 were 2.5 and 1.7 in Con and AM at Be-EMF, respectively, and 4.1 and 5.2 in Con and AM at Af-EMF, respectively. The numbers of bouts of Day 6 are smaller at Be-EMF than Af-EMF, which trend is similar of Day 4. I do not understand the meaning of “opposite”. Line 232-234: What do you want to say in this sentence? Line 239 and 253: The “P>0.05” is not correct, but “P>0.1”. Fig 1 and 2: Means of bars should be written in margins in these figures: SD or SEM. Line245: It must be written that duration and bouts of “Jump-running” are significantly larger in AM than Con. Line274: It is too extrema to add “walking behavior” in this sentence, which did not increased in AM except Table 2.
Reviewer 2 Report
The paper is generally well written and well presented. The study is interesting and very topical. There has been similar work conducted previously. The authors should acknowledge the previous studies in both the introduction and in the discussion. Examples of studies that show the effects of early calf handling on behavioural responses include:
Krohn, C.C., Jago, J.G. and Boivin, X., 2001. The effect of early handling on the socialisation of young calves to humans. Applied Animal Behaviour Science, 74(2), pp.121-133.
Jago, J.G., Krohn, C.C. and Matthews, L.R., 1999. The influence of feeding and handling on the development of the human–animal interactions in young cattle. Applied Animal Behaviour Science, 62(2-3), pp.137-151.
Charlton, G.L. and Bleach, E.C.L., 2007, April. Responses of positively handled calves to human interactions and disbudding. In Proceedings of the British Society of Animal Science (Vol. 2007, pp. 43-43). Cambridge University Press.
The conclusions should more succinctly reflect the findings of the present study.
Some minor revisions:
Introduction
Line 36 – replace ‘ for the sake of fetal membrane removal’ with ‘to remove fetal membranes’
Line 56 – delete ‘are’
Line 57 - delete ‘breeding stock’
Line 58 - delete ‘ stocking’
Line 59 - replace ‘whereas’ with ‘however’; replace ‘those’ with ‘the’; replace ‘amounts’ with ‘number’
Line 62 - delete ‘the’
Materials and methods
Line 86 – replace ‘allantoic’ with ‘allantois’
Line 88 - state the mean BCS of the cows
Line 90 - state reference for calving ease score
Line 92 – replace ‘right’ with ‘ immediately’
Line 94 insert ‘for’ in front of ‘6 to 12 hours; delete ’later’
Line 95 – lowercase ‘h’ for hutch
Line 98 – what was colostrum quality?
Line 102 – insert ‘open’ before ‘bucket’
Line 102 – what type of milk was fed? Was it whole cow’s milk or a calf milk replacer? If the latter, what was the concentration and nutritional value of the milk feed?
Line 103 – What type of starter feed was offered? Pellets or coarse mix? Was forage (straw or hay) offered too? Was water available for the calves?
Line 107 – replace ‘amounts’ with ‘numbers’
Line 110 – Replace ‘right’ with ‘from immediately’
Line 114 – Replace ‘the AM calves were licking mimicked by manual brush grooming’ ‘licking of the AM calves by their dam was mimicked by manually grooming with a brush’
Line 114 – Replace ‘born’ with ‘ birth’
Line 119 - Delete ‘performance’
Line 122 – Replace ‘born’ with ‘ birth’
Line 128 – replace ‘not being’ with ‘were not’
Line 125 – lowercase ‘h’ for hutch
Table 1 – Object S/L – replace ‘tough’ with ‘tongue’
Object sniffing – insert ‘contact’ after ‘close’
Object sniffing – insert ‘contact’ after ‘close’
Human sniffing – insert ‘contact’ after ‘close’
Human R/L – replace ‘tough’ with ‘tongue’
Line 172 – Replace ‘We analysed all data ……’ with ‘All data were analysed ….’
Line 227 – replace ‘while’ with ‘ as well as’
Line 231 – 232 – Check that the statement (or the data) is correct. The data in Table 3 suggest self-grooming is also higher after feeding on day 6.
Line 233 - Replace ‘period of’ with ‘behaviours’
Line 242 – Figure 1 – What do the error bars represent?
Line 356 – Figure 2 – Define ‘Human R/L’
Discussion
Acknowledge the findings of the studies highlighted above.
Reviewer 3 Report
This study set out to investigate the effects of what they refer to as artificial mothering on activity and human affinity in Holstein calves. The calves were immediately removed from the dam and then individually housed; the control calves had limited contact with the stockpersons whereas the treatment calves were brushed for 30 min immediately after birth and then 5 min twice daily for the next week.
I would argue that the treatment applied does not mimic all aspects of maternal behavior in the hours following birth and thus recommend that the title be changed accordingly. Suggested title would be “the effect of short term brushing immediately after birth on dairy calf activity in the week after birth.”
Also, if I read this correctly the authors did not control for presence of the human during the brushing of the calves across treatments. For instance, one could argue that the increased activity of the AM calves could simply be a consequence of these calves being disrupted by the human. For this to be controlled the control calves should have also been disrupted but not groomed – was this done?
Lastly, the authors have run numerous tests but do not provide any predictions as to what they believed would change in response to their treatment. This should be well thought out in the introduction and then only those behaviors that are predicted to change be tested. This would dramatically reduce the result sections which primarily shows NS findings.
Line 42 – these are perceived benefits – please see also the two systematic reviews published in 2019 on this topic in the Journal of Dairy Science
Line 81-93 – there is a considerably body of work indicating that dystocia can affect calf vitality – please ensure that treatment groups were balanced for this factor
Line 100-101 – feeding 6 L per day is known to create hunger in calves (de Paula Vieira et al., 2008) and therefore this must be acknowledged given that it also increases oral behavior in calves. Also, please note that there is a large body of evidence indicating that calves are not able to consume significant amounts of starter before they are 14-21 days of age and thus I would predict that this finding is reflective of head in bucket rather than intake per se. Also do the authors have a prediction for this – if not then please remove.
Figure 3 – I find this figure difficult to interpret as it is well known that calves less than about 3 weeks of age only consume negligible amounts of starter. I suggest that this figure refer to time spend with head in bucket rather than intake.
Table 2 – it is not clear what the units are in this table? I am assuming that these are minutes?
Line 283 – I would argue that the increased oral behavior is not necessarily reflective of increased appetite but rather a consequence of reduced fear. There is considerable evidence that individually housed calves experience higher levels of fear than group housed calves. It could therefore be that the presence of the human twice daily may have reduced the fear response in these calves thereby allowing them to engage in highly motivated behaviours – which in these restricted milk intake calves could manifest in increased oral behaviours.
Line 292 – 302 – I argue that there is insufficient evidence to speculate that the calves in the AM group are indeed have an increased appetite. The study did not control for the presence of the human and the downstream effects on fear. Please remove these speculative comments.
Line 304-305 – again there is a body of new evidence indicated that calves are highly motivated to suck and ingestion of solid feed does not reduce this motivation (see Lindfors et al and others on this topic).
Line 337 – please rephrase “good human” as this is vague
Conclusions – I suggest that the authors focus their study on the measures which they predict will be different. Also, they must also consider that these calves were fed restricted amounts of milk and thus were hungry. A calf provided adequate amounts of milk during the first week of life may behave differently.
References – please refrain from citing books or other non-peer reviewed documents (references, 5, 17, 18)
Round 2
Reviewer 3 Report
I am still not in agreement with the study title - can you please ask them to exchange artificial mothering with grooming.
Author Response
Dear reviewer,
Thank you for your comments. After consideration, the authors agree with your suggestion. The "artificial mothering" was replaced by "artificial grooming" in the whole manuscript. Now it feels more appropriate. Thank you again.
Best regards
Congcong Li